# The Adaptive Mechanisms and Checkpoint Responses to a Stressed DNA Replication Fork

**DOI:** 10.3390/ijms241310488

**Published:** 2023-06-22

**Authors:** Joanne Saldanha, Julie Rageul, Jinal A. Patel, Hyungjin Kim

**Affiliations:** 1The Graduate Program in Genetics, Stony Brook University, Stony Brook, NY 11794, USA; 2Department of Pharmacological Sciences, Stony Brook University, Stony Brook, NY 11794, USA; 3Stony Brook Cancer Center, Renaissance School of Medicine, Stony Brook University, Stony Brook, NY 11794, USA

**Keywords:** DNA replication, genome stability, ATR-CHK1, fork protection complex, replication stress response, DNA damage tolerance

## Abstract

DNA replication is a tightly controlled process that ensures the faithful duplication of the genome. However, DNA damage arising from both endogenous and exogenous assaults gives rise to DNA replication stress associated with replication fork slowing or stalling. Therefore, protecting the stressed fork while prompting its recovery to complete DNA replication is critical for safeguarding genomic integrity and cell survival. Specifically, the plasticity of the replication fork in engaging distinct DNA damage tolerance mechanisms, including fork reversal, repriming, and translesion DNA synthesis, enables cells to overcome a variety of replication obstacles. Furthermore, stretches of single-stranded DNA generated upon fork stalling trigger the activation of the ATR kinase, which coordinates the cellular responses to replication stress by stabilizing the replication fork, promoting DNA repair, and controlling cell cycle and replication origin firing. Deregulation of the ATR checkpoint and aberrant levels of chronic replication stress is a common characteristic of cancer and a point of vulnerability being exploited in cancer therapy. Here, we discuss the various adaptive responses of a replication fork to replication stress and the roles of ATR signaling that bring fork stabilization mechanisms together. We also review how this knowledge is being harnessed for the development of checkpoint inhibitors to trigger the replication catastrophe of cancer cells.

## 1. Initiation of DNA Replication

DNA replication is fundamental to the maintenance and diversification of life. In eukaryotic cells, DNA replication initiates from multiple origins deployed across the whole genome, implying the need for a tight orchestration of their firing. Complex multi-step regulatory mechanisms coordinate such efforts and ensure that the genome is fully duplicated. Notably, a large excess of DNA replication origins are present throughout the human genome, with only 5–10% of them firing throughout S phase [1]. Based on their usage, DNA replication origins are classified into the following three categories: (1) constitutive origins that invariably fire in all cells of a population, (2) flexible origins (the majority) that only fire in some cells of a population, and (3) dormant origins that are kept silent during normal conditions but can become activated upon DNA damage when a replication fork stalls in the vicinity [2,3].

Eukaryotic DNA replication initiation follows a conserved process that leads to the assembly of two replisomes, which will move away from the activated origin in opposite directions [4]. In budding yeast, at the core of the replisome is a hexameric DNA helicase motor, composed of the minichromosome maintenance subunits 2 to 7 (MCM2-7), in the shape of a ring that unwinds parental double-stranded DNA (dsDNA) [5]. To avoid genome re-replication and thus strictly limit replication to once per S phase, the activation of DNA replication origins entails a two-step process consisting of (1) DNA helicase loading onto DNA and (2) DNA helicase activation, which occurs at temporally distinct times. The first step takes place in the G1 phase via the “licensing” of all replication origins, rendering them competent for firing. The second step occurs in the early S phase via the activation of the DNA replicative helicase and the formation of two active replication forks through the regulated sequential recruitment of firing factors (Figure 1). The temporality of this two-step process is governed by the activity of global cell cycle regulators, mainly the Ser/Thr protein kinases CDKs (cyclin-dependent kinases) and DDK (Dbf4-dependent kinase, also known as CDC7-DBF4), ensuring that the DNA replication program is coupled with cell growth conditions and environmental cues [6]. A recent study also demonstrated that CDK1 plays a redundant role with CDC7 in the G1/S transition [7].

### 1.1. Origin Licensing: Pre-RC Formation in G1 Phase

DNA replication origins are first recognized by the heterohexameric origin recognition complex (ORC1-6) that possesses DNA-binding domains and exhibits AAA^+^ ATPase activity. In mammals, ORC1 and ORC6 weakly interact with the ORC2-5 core, thus forming a dynamic open ring that binds to DNA through its central cavity [8]. ORC1 is loaded onto DNA as early as in mitosis and is degraded during the S phase through the ubiquitin-proteasome system, suggesting that it promotes the timely assembly of the ORC complex at its origins [9]. In yeast, Orc1-6 binding to origins induces a small bend in dsDNA followed by the binding of another ATPase, Cdc6, thus closing the Orc ring around duplex DNA [10]. Then, CDT1 (CDC10-dependent transcript 1) is recruited to ORC1-6/CDC6 bound origins, forming the platform necessary to recruit and sequentially load two MCM2-7 hexamers in a head-to-head orientation using ATP hydrolysis. In yeast, Cdt1 forms a complex with MCM2-7 before being loaded onto origins [11]. At this stage, the MCM double hexamer is inactive as it encircles duplex DNA. Together, this assembly of proteins forms the pre-replication complex (pre-RC) at the licensed replication origins. Importantly, helicase loading can only occur in the G1 phase during low CDK activity because of the inhibitory effect of the direct phosphorylation of ORC, CDC6 or MCMs [12,13,14]. The CRL4^CDT2^ ubiquitin E3 ligase targets CDT1 for degradation in a proliferation cell nuclear antigen (PCNA)-dependent manner to prevent DNA re-replication [15]. Additionally, CDT1 is inhibited through sequestration by the S phase-specific inhibitor GEMININ, thus preventing pre-RC assembly [16].

### 1.2. DNA Helicase Activation: Pre-IC Formation at the G1-S Phase Transition

The activation of the replicative DNA helicase requires a series of allosteric changes that only occur during high CDK activity in the S phase, allowing for the phosphorylation and recruitment of firing factors. Importantly, DDK transiently binds to the pre-RC and phosphorylates key sites on the double hexamer, most notably on the MCM4 and MCM6 subunits, which induces structural changes that allow for the binding of CDC45 (cell division cycle 45) and GINS heterotetramer (Go-Ichi-Ni-San, meaning 5-1-2-3 in Japanese) composed of Sld5 (synthetic lethal with Dpb11), Psf1 (partner of Sld5-1), Psf2, and Psf3 [17,18,19]. Simultaneously, additional phosphorylation events by CDKs and DDK of critical firing factors, such as TRESLIN, ATP-dependent DNA helicase Q4 (RECQL4; Sld3 and Sld2 in *S. pombe*), DNA topoisomerase 2-binding protein 1 (TOPBP1), and DNA polymerase ε (pol ε), collaboratively contribute to the assembly of the pre-initiation (pre-IC) complex with two replicative helicases formed by the tight CMG (CDC45-MCM2-7-GINS) complex at its core [20,21,22,23,24]. A recent systemic chromatin immunoprecipitation from budding yeast revealed distinct intermediates of the pre-IC assembly that are mutually dependent on the origin firing factors [25].

### 1.3. Origin Firing: Formation of Two Functional DNA Replication Forks

The CMG formation results in the release of ADP from the MCM double hexamer, which allows binding of ATP, thus triggering the initial untwisting of a short stretch of DNA and the separation of two inactive CMG complexes. MCM10 and ATP hydrolysis are instrumental for helicase activity, such that the two replisomes pass each other by translocating 3′ to 5′ on the leading strand template (with their N-terminus at the front of the helicase) after evicting the lagging strand template from the MCM pore [26,27]. Replication protein A (RPA) quickly binds to the exposed single-stranded DNA (ssDNA) after DNA melting, while CDC6 and CDT1 are evicted and inactivated. DNA polymerases α and δ are then recruited along with replication factor C (RFC) and PCNA to convert the pre-IC into two active replication forks, which move in opposite directions from the activated origin. DNA is thus replicated in a semi-conservative fashion, while deoxyribonucleoside triphosphates (dNTPs) are incorporated at daughter strands, in which the leading strand is copied continuously by DNA Pol ε, while the lagging strand is copied discontinuously as a succession of Okazaki fragments initiated by DNA Pol α primase (RNA primer) and elongated by DNA Pol δ [4].

## 2. Control of DNA Replication by the Fork Protection Complex

The activities of DNA unwinding and synthesis in the replisome need to be tightly coupled; otherwise, uncoupling of replisome activity results in a large stretch of ssDNA accumulation that can lead to fork breakage and collapse, which is considered a major source of chromosome aberrations and instability. The architecture of the replication fork is primarily maintained by the fork protection complex (FPC), which physically interacts with the replication machinery and tethers the CMG helicase and replicative polymerase activities to prevent uncoupling of DNA replication, and thus limit ssDNA exposure [28]. The FPC is composed of a heterodimeric complex of TIMELESS (TIM) and TIPIN (Tof1 and Csm3 in *S. cerevisiae*; Swi1 and Swi3 in *S. pombe*), as well as CLASPIN/Mrc1 and AND-1/Ctf4 [29] (Figure 2). Several replisome-associated proteins, including poly(ADP-ribose) polymerase 1 (PARP1), DDX11 helicase, and silencing defective 2 (SDE2), are known to directly interact with the FPC via TIM [30,31,32,33]. Loss of FPC components results in a decrease in replication fork speed, accumulation of ssDNA, and frequent fork stalling, suggesting that the FPC is an essential scaffold necessary for the integrity and function of the replisome [32,34,35,36,37]. Recently, detailed views of both yeast and human replisomes were revealed by cryo-electron microscopy, which provides important insights into the structure and function of the FPC in modulating replisome activity [38,39]. Somewhat counterintuitively, the TIM-TIPIN complex is localized at the front of the CMG and grips the parental DNA duplex before strand separation, rather than positioned between the CMG and replicative polymerases to physically tether them. The extensive interactions between TIM and the CMG, including the N-terminus of MCM6 that extends into the core of TIM, stably place the FPC ahead of the replisome. At the same time, a positively charged groove of the TIM-TIPIN complex latches onto a complete turn of dsDNA in front of the fork junction. This helps arrange DNA contacts with the MCM complex in a way that facilitates the separation of a lagging strand template that is forced out from the MCM central pore [39]. Therefore, the FPC is necessary for the precise positioning of the CMG to facilitate fork progression and constitutes part of an important DNA unwinding mechanism for strand separation.

A series of biochemical studies using recombinant replisome constituents indicated that the FPC directly controls the rate of replication fork progression. Earlier biochemical studies revealed that the purified TIM-TIPIN complex directly interacts with the MCM subunits and inhibits the DNA unwinding and ATPase activities of the CMG, while it stimulates the activities of polymerases α, δ, and ε, indicating that the FPC is able to affect the catalytic activities of the replisome [40]. Recently, both yeast and human replisome proteins were reconstituted in vitro to further support these findings [41,42]. In both cases, while the rate of DNA synthesis was low with the minimally reconstituted replisome, the addition of purified *S. cerevisiae* Mrc1-Tof1-Csm3 or human CLASPIN-TIM-TIPIN complexes greatly accelerated replication fork progression, suggesting that the FPC stimulates the activity of the replisome. It is notable to observe that CLASPIN/Mrc1 alone is able to promote replication fork elongation in the absence of TIM-TIPIN/Tof1-Csm3, whereas TIM-TIPIN/Tof1-Csm3 themselves fail to do so, indicating that one primary role of TIM-TIPIN is to augment the activity of CLASPIN. As TIM and CLASPIN directly interact, this suggests that the ability of TIM to grip dsDNA and hold the CMG may stabilize the association of CLASPIN to the replication fork, thereby maximizing DNA synthesis. Therefore, the FPC is necessary for efficient replisome progression by exerting a scaffold role in maintaining the integrity of the replisome, while regulating the catalytic activity of DNA unwinding and synthesis.

On the other hand, it is also known that the FPC restricts the replication process by pausing the replisome at various fork barriers during normal fork progression and under replication damage. An early study in *S. cerevisiae* revealed that the FPC acts as a replication-pausing complex in response to fork arrest induced by hydroxyurea (HU) to prevent the uncoupling of the CMG activity from DNA synthesis [43]. Accordingly, Tof1 or Mrc1 mutations are associated with a temporal increase in ssDNA accumulation and activation of the S phase checkpoint. The positioning of TIM/Tof1 ahead of the CMG further corroborates the idea that one important role of TIM is likely to hold the CMG and restrict its uncontrolled unwinding activity, thereby keeping the replisome coupled and suppressing extensive ssDNA exposure. Within the FPC, both the replication-promoting function of CLASPIN and the restrictive function of TIM may fine-tune the rate of replication fork progression and temporarily pause in response to various obstacles in order for cells to activate the replication checkpoint and stabilize stalled forks (as discussed later).

That said, the FPC is known to be necessary for replisome progression through distinct fork barriers composed of both proteins and DNA elements. For instance, the individual constituents of the FPC, including TIM/Tof1, TIPIN/Csm3, or CLASPIN/Mrc1, are required for navigating through intrinsically difficult-to-replicate DNA regions such as trinucleotide repeats, telomeres, and guanine (G)-quadruplex DNA structures that are prone to form secondary structures [44,45]. TIM was shown to directly recognize the G-quadruplex structure and recruit the DDX11/Chl1 helicase to resolve secondary DNA structures [46,47]. Additionally, the programmed sites in *S. pombe* marked by protein-DNA interacting barriers near the mating-type (*mat1*) locus or the rDNA locus in *S. cerevisiae* are other examples where transient pausing activity of the FPC is involved [48,49]. Furthermore, irreversible protein-DNA crosslinks such as the one aberrantly formed by topoisomerase poisons (e.g., camptothecin and irinotecan) are an important barrier that the FPC needs to deal with. The in vitro reconstitution system revealed that Tof1-Csm3 is necessary and sufficient for pausing at topoisomerase I (TopI)-induced replication fork barriers in an orientation-dependent manner [50]. As originally identified as a TopI-binding factor [51], TIM/Tof1 may couple replisome and TopI activities by directly recruiting TopI ahead of the CMG, which would allow for sensing and resolving protein-DNA lesions to relieve torsional stress, and thus facilitate fork progression.

## 3. Fates and Responses of a Stalled DNA Replication Fork

Stalled replication forks show a remarkable degree of plasticity to adapt to replication stress, thereby readily being able to restore fork integrity and restart DNA synthesis. DNA damage tolerance (DDT) is a mechanism that enables cells to continue replicating their DNA even in the presence of damage to prevent persistent fork stalling, which includes fork reversal as part of template switching (TS), repriming, and translesion DNA synthesis (TLS) (Figure 3). Specifically, overcoming DNA lesions while minimizing the effect on fork progression is critical for the timely completion of DNA replication.

### 3.1. Fork Reversal

Fork reversal is a mechanism of fork stabilization that is frequently utilized when DNA replication forks encounter DNA damage or lesions, allowing stalled forks to reverse their course until the damage is repaired, before normal replication can be restarted [52]. In this process, a typical three-way junction of the replication fork is converted into a four-way junction by the coordinated annealing of two nascent DNA strands behind the fork to form a regressed fork or “chicken foot structure”. Lesions on the template strand are either repaired or bypassed in an error-free manner using a homologous template via TS, usually the nascent daughter strand on the sister chromatid. The RAD51 recombinase plays two distinct roles in this process; first, RAD51 creates a paranemic DNA duplex (i.e., side-by-side DNA strands that are separable without the need to rotate the opposite strand) behind the CMG via its strand exchange activity, which is used as a substrate by DNA translocases for branch migration to catalyze fork reversal [53]. This allows the CMG poised ahead of a reversed fork to readily restart DNA synthesis. Then, formation of stable RAD51 nucleofilaments mediated by BRCA2 onto ssDNA protects the reversed fork from nucleolytic cleavage [54].

The fork reversal process is at least partly regulated by SNF2-family DNA annealing helicases, including SMARCAL1 (SWI/SNF-related, matrix-associated, actin-dependent, regulator of chromatin, and subfamily A-like 1), ZRANB3 (zinc finger, RAN-binding domain containing 3), and HLTF (helicase-like transcription factor), which rewind two DNA strands in an ATP-dependent manner. The HARP-like domain of both SMARCAL1 and ZRANB3 is required for annealing helicase activity, but not for DNA binding and ATPase activity [55]. SMARCAL1 is recruited to stalled forks via its interaction with RPA32 [56,57]. The polarity of RPA on the leading and lagging strands is critical for the regulation of SMARCAL1 activity, with RPA being inhibitory on the lagging strand while facilitating SMARCAL1 activity on the leading strand [58]. ZRANB3 does not interact directly with RPA but rather recognizes polyubiquitinated PCNA through its PIP (PCNA-interacting protein)-box motif, which together with its APIM (AlkB homology 2 PCNA-interaction motif) is critical for the localization of ZRANB3 to DNA damage sites [59]. The polyubiquitination of PCNA at Lys63 is carried out through the combined activity of the RAD6-RAD18 E2-E3 ubiquitin ligase complex and HLTF and SHPRH (SNF2 histone linker PHD RING helicase) [60]. HLTF is a DNA-dependent ubiquitin E3 ligase that catalyzes the formation of polyubiquitin chains on PCNA by first transferring the ubiquitin moiety onto the RAD6-Ub in complex with RAD18 [61]. In a reaction catalyzed by RAD18, the thiol-linked ubiquitin chain on RAD6 is then transferred to unmodified PCNA. Additionally, the highly conserved HIRAN (HIP116/HLTF Rad5 N-terminus) domain of HLTF captures free 3′-hydroxyl group on the nascent leading strand, directing HLTF-mediated fork reversal [62]. As all three enzymes catalyze similar fork remodeling in vitro, it is unclear why the three co-exist in the cells. Since there are additional functionalities that are present in ZRANB3 (i.e., a nuclease domain) and HLTF (i.e., a ubiquitin ligase domain), one can expect that they may be required for dealing with distinct fork structures generated by different types of replication stress. Perhaps, distinct substrate specificities and recruitment strategies may be necessary for mediating the sequential steps of fork remodeling and adapting to variable DNA repair intermediates.

A regressed arm at the reversed fork undergoes controlled resection by specific nucleases to enable efficient fork restart. Conversely, deregulation of fork processing and protection leads to extensive degradation of the nascent strands. MRE11, a DNA endonuclease and 3′-5′ exonuclease, is recruited to stalled forks by PARP1 to facilitate fork resection and restart [63]. As BRCA2 stabilizes the RAD51 nucleofilaments on the regressed DNA to protect it, BRCA2-deficient cells exhibit extensive nascent strand degradation not only by MRE11 but also by other nucleases, including CtIP and EXO1, which initiate and extend MRE11-mediated degradation [64]. RAD52, together with PTIP and MLL4, is also required for recruiting MRE11 and priming MRE11-dependent fork resection in BRCA2-deficient backgrounds [65]. Several factors that protect reversed forks from nucleolytic degradation have been noted. The Fanconi anemia (FA) protein FANCD2 prevents MRE11-mediated fork processing by stabilizing RAD51 nucleofilaments similarly to BRCA2 [66]. BOD1L restrains nucleolytic degradation by inhibiting BLM and FBH1 helicases, and blocking DNA2-mediated resection [67]. BOD1L interacts with SETD1, which catalyzes histone H3K4 methylation at replication forks, and thus enhances FANCD2-dependent histone chaperone activity [68]. Additionally, ABRO1 prevents DNA2/WRN-dependent degradation of stalled forks, which is distinct from the BRCA2-dependent pathway that prevents MRE11-mediated fork degradation [69].

Following repair, reversed forks are subject to restart either by RECQ1-mediated branch migration or DNA2/WRN-mediated DNA unwinding and resection [70,71]. The WRN ATPase promotes DNA2 activity to process a reversed fork with a 5′-3′ polarity, while RECQ1 restricts DNA2-mediated end resection [71]. RECQ1 independently restarts a reversed fork via its branch migration activity, and poly(ADP-ribosyl)ation by PARP1 inhibits RECQ1 activity to prevent premature fork restart [70]. In addition, structure-specific nucleases process reversed forks via incision of the fork junction upon prolonged fork stalling. For instance, loss of BRCA2 causes fork breakage and one-ended double strand break (DSB) formation, which must then be resolved through the engagement of a specialized break-induced repair (BIR) pathway carried out by MUS81 and SLX4 to process this intermediate structure and allow for fork restart [64]. BIR is a specialized subtype of homologous recombination (HR) that involves 5′ to 3′ end-resection, thus generating a 3′-ssDNA-RPA coated end. This is followed by MRE11/CtIP/EXO1-mediated end resection, RAD52-mediated strand invasion, and POLD3-dependent DNA synthesis, which results in the formation of intermediates that are processed by MUS81/SLX4 [72].

### 3.2. Fork Restart by Repriming

Repriming occurs at stalled forks to re-initiate DNA synthesis beyond a DNA lesion, which is particularly relevant during continued leading strand synthesis. Identification of the enzymatic activity catalyzed by DNA-directed primase/polymerase PRIMPOL in human cells suggests that repriming constitutes an alternative mechanism for damage tolerance and fork restart conserved in higher eukaryotes [73,74,75]. PRIMPOL belongs to the archaea–eukaryotic primases (AEP) superfamily and exhibits both nuclear and mitochondrial localization in human cells [73]. In response to structural hindrances such as UV lesions, G-quadruplexes, chain terminating nucleosides, and R-loops that result in replication blockage, PRIMPOL mediates fork restart downstream of the lesion via RPA binding and through its DNA primase activity to generate a DNA primer [75]. Its polymerase activity extends the DNA primer, thereby skipping the lesion; however, this leaves behind ssDNA gaps that need to be filled in a post-replicative manner.

PRIMPOL can directly read lesions such as 8-oxo-dG in vitro; however, this has not yet been demonstrated in vivo [76]. The preference of PRIMPOL in the incorporation of dNTPs can be attributed to the presence of Tyr100 at its 3′-elongation site, which sterically hinders the 2′-hydroxyl group of NTPs, while favoring the binding of dNTPs. Interestingly, a cancer-associated missense mutation, Y100H, results in preferential incorporation of NTPs over dNTPs at sites of DNA damage, generating an RNA:DNA hybrid structure. Furthermore, overexpression of the Y100H mutant in PRIMPOL-deficient cells leads to an increased tolerance to HU [77]. Resistance to HU, methyl methanesulfonate (MMS), and mitomycin C (MMC) is observed with heightened PRIMPOL activity, suggesting that repriming can alleviate replication stress [78]. Emerging evidence supports the existence of diverse regulatory mechanisms to control PRIMPOL activity and repriming. PRIMPOL is phosphorylated by Polo-like kinase 1 (PLK1) at its RPA binding motifs, which is differentially modified throughout the cell cycle and prevents aberrant engagement of PRIMPOL to chromatin [79]. PolDIP2 interacts with PRIMPOL to stimulate its polymerase activity [80], and USP36 removes Lys29-linked polyubiquitination of PRIMPOL, thereby increasing its cellular levels [81]. In addition, BRCA2 associates with MCM10 to suppress PRIMPOL-mediated repriming; accordingly, BRCA2-deficient cells fail to restrain fork progression due to increased repriming, leaving behind ssDNA gaps [82]. Similarly, loss of TLS Pol ι unleashes PRIMPOL-mediate repriming, which accelerates nascent DNA elongation and reduces the S phase checkpoint activation, causing genome instability in the M phase [83]. Furthermore, in heterozygous *Brca2* mouse cells, defective RAD51 stabilization results in increased PRIMPOL activity and ssDNA gaps, which renders cells hypersensitive to 5-(hydroxymethyl)-2′-deoxyuridine (5-HmdU) due to the formation of abasic sites by SMUG1 glycosylase [84]. Aberrant engagement of PRIMPOL is expected to cause ssDNA gap accumulation and unrestrained fork progression, representing a potential replication vulnerability of cancer cells. Oncogene activation may exacerbate this process; for instance, the MDM2 oncoprotein was shown to ubiquitinate and destabilize PARP1, thereby promoting PRIMPOL-dependent repriming [85].

### 3.3. Translesion DNA Synthesis (TLS)

TLS is a type of DDT mechanism that involves the engagement of low-fidelity polymerases to bypass a DNA lesion. Unlike the high-fidelity B-family replicative polymerases Pol α, Pol δ and Pol ε, the Y-family TLS polymerases, such as Pol η, Pol κ, Pol ι and Rev1, lack proof reading ability and possess lower processivity. While the two families of DNA polymerases share common structural features such as a ‘fingers’, ‘thumb’ and ‘palm’ domains, Y-family polymerases also possess an additional ‘little finger’ domain that allows additional flexibility of their active sites, granting these polymerases with the ability to accommodate bulky, distorted bases, further contributing to base pair mismatching [86]. As a result, TLS engagement can contribute to increased mutagenesis and tumorigenesis.

By nature, TLS is a useful strategy to achieve undisrupted and timely completion of DNA replication, especially by resolving the uncoupling of helicase-polymerase activities caused by a replication-blocking lesion, such as UV-induced cyclobutane pyrimidine dimers (CPDs). The engagement of TLS to a DNA lesion requires polymerase switching, during which replicative polymerases are replaced by specific TLS polymerases. TLS polymerases are recruited by monoubiquitination of PCNA at Lys164 catalyzed by the RAD6-RAD18 E2-E3 ubiquitin ligase complex, and many TLS polymerases harbor specialized ubiquitin-binding motifs that specifically recognize monoubiquitinated PCNA along with their PIP box motifs for their interaction with PCNA [87]. A recent in vitro reconstitution of TLS Pol η to the eukaryotic replisome demonstrated that Pol η facilitates “on-the-fly” bypass of a leading-strand CPD lesion to rapidly restart uncoupled replication forks, and monoubiquitinated PCNA stimulates this process by outcompeting Pol δ that inhibits TLS [88]. For the lagging strand CPD, ssDNA gaps and stalled Okazaki fragments are accumulated, in which Pol η was shown to promote TLS in a gap-filling manner. A high number of DNA lesions and extensive ssDNA generation is likely to promote repriming and gap-filling process in comparison to the on-the-fly mode, whose activity may be coupled with PRIMPOL. In line with this, two temporally distinct pathways involved in filling ssDNA gaps were observed. Specifically, a TLS mechanism dependent on PCNA monoubiquitin and the REV1-Pol ζ complex fills gaps in the G2 phase, whereas the E2 conjugating enzyme UBC13, RAD51, and REV1-Pol ζ are responsible for gap filling in the S phase [89]. In both cases, BRCA1/2 promote gap filling processes by restricting MRE11 activity.

Ubiquitin-specific protease 1 (USP1) is a deubiquitinase that removes ubiquitin from PCNA and FANCD2. While USP1 constitutively deubiquitinates PCNA to prevent the abnormal engagement of TLS polymerases, notably Pol κ, USP1 undergoes its own autocleavage and degradation upon UV damage, allowing for PCNA monoubiquitination to be elevated [90,91]. While the USP1 autocleavage mutant is still able to deubiquitinate PCNA, its expression results in increased fork stalling and premature fork termination. These replication defects result from defective USP1 recycling and aberrant USP1 trapping to DNA, owing to its failure to be removed by the metalloprotease SPRTN [92]. Loss of USP1 causes synthetic lethality with BRCA1/2 deficiency due to aberrant processing of PCNA ubiquitination and engagement of TLS polymerases, leading to ssDNA gap accumulation, which synergizes with the toxicity of PARP inhibition [93].

### 3.4. Balancing the Three Acts

Fine-tuning the balance of distinctive DDT pathways to effectively overcome replication obstacles is critical for ensuring timely DNA replication and protecting replication fork integrity. There are several factors that would dictate the choice underlying how cells respond to replication damage. First, the DDT pathway choice is largely regulated by the nature and location of the DNA damage. For example, while UVC-induced CPDs on the leading strand are readily bypassed by TLS mediated by Pol η, more distorted pyrimidine-pyrimidone (6-4) photoproduct (6-4PP) lesions favor repriming, as evidenced by ssDNA gap accumulation behind replication forks [94,95]. Stalled forks accumulated by multiple doses of cisplatin are bypassed by PRIMPOL-mediated repriming that is upregulated by the replication stress response [96]. It was also shown that traversing DNA inter-strand crosslinks (ICLs), an absolute roadblock of replication, requires repriming downstream of the ICL lesion, which is mediated by PRIMPOL and is repaired post-replicatively [97].

Second, the extent of fork stalling and the genetic background of cells determine preferences. While BRCA1/2 deficiency results in nucleolytic degradation of reversed forks at a high dose of HU, the treatment of mild or low HU concentration in BRCA1/2-deficient cells favors repriming associated with ssDNA gap formation [98]. Suppression of fork reversal by the loss of fork remodelers such as SMARCAL1 and HLTF results in PRIMPOL-dependent accumulation of ssDNA gaps under replication damage, including cisplatin and HU, implicating a preference toward repriming for fork rescue [96,99]. PARP inhibition also reduces fork reversal and upregulates ssDNA gap formation by repriming [96,100]. RAD51 depletion leads to persistent DNA synthesis across UV-damaged DNA via MRE11 and PRIMPOL, further supporting that the inhibition of fork reversal favors repriming [101]. Interestingly, while HLTF knockout cells exhibit unrestrained fork progression associated with PRIMPOL-dependent repriming, the HIRAN domain mutant relies on REV1-mediated TLS activity for fork progression and mitigation of replication stress [99]. Similarly, the FANCJ helicase S990A mutant is TLS-prone, exhibiting unrestrained fork elongation but without ssDNA gap accumulation or fork reversal [102,103]. Together, these findings provide compelling evidence that certain genetic backgrounds can influence the choice of DDT pathways. The formation of distinct DNA structures and unique levels of associated DNA damage caused by the mutant enzymes may be responsible for fine-tuning regulation for the pathway choice.

Third, post-translational modifications of PCNA represent yet another regulatory mechanism in DDT pathway choice. Specifically, RAD18-dependent PCNA monoubiquitination is critical for TLS engagement while PCNA polyubiquitination promotes TS, including fork reversal, where HLTF acts upstream of ZRANB3 that is recruited to polyubiquitin of PCNA [104]. Silencing RAD18 or Pol κ suppresses the ssDNA gaps manifested in USP1-deficient cells and rescues the synthetic lethality between USP1 and BRCA1, suggesting that the aberrant engagement of TLS is related to gap accumulation [93]. Co-deficiency of PRIMPOL and Pol η causes pronounced fork stalling and a hypersensitivity to UV damage, indicating that repriming and TLS could be complementary [105]. PRIMPOL may be required for efficient fork progression under unchallenged conditions, whose activity is expected to cooperate with TLS for efficient DNA damage tolerance.

## 4. Fork Stabilization Mechanisms by the ATR Checkpoint

### 4.1. Activation of the ATR Checkpoint

Stalling of DNA replication forks triggers the activation of the ataxia telangiectasia and Rad3-related (ATR) checkpoint to stabilize stalled forks and promote their recovery. A common DNA structure that activates the ATR checkpoint is ssDNA at stalled forks, mostly generated by uncoupling of the CMG and replicative polymerase activities in the replisome [106]. This may not necessarily involve a physical separation of helicase and polymerases, especially in a situation where the replisome is still tethered together, but polymerases cannot proceed due to the presence of a physical lesion and the CMG keeps unwinding the parental DNA duplex, which is likely to generate stretches of ssDNA. Binding of RPA to ssDNA serves as a platform to recruit ATR and its associated regulatory protein ATRIP, which promotes localization of ATR to sites of DNA damage [107]. However, recruitment to RPA-coated ssDNA alone is not sufficient for activating ATR; ATR activation relies on at least two ATR activators in vertebrates, topoisomerase II binding protein (TOPBP1) and Ewing tumor-associated antigen (ETAA1), both of which directly stimulate ATR kinase activity through their conserved ATR-activation domains (AADs) [108,109]. TOPBP1 is recruited specifically to a free 5′-ended ssDNA-dsDNA junction, to which the MRE11-RAD50-NBS1 (MRN) complex and the RAD9-RAD1-HUS1 (9-1-1) checkpoint clamp are loaded to promote the recruitment of TOPBP1 [110,111]. A 5′-ended junction with an RNA-DNA primer can be rapidly accumulated on the discontinued lagging strand upon fork stalling, especially when the supply of dNTPs is limited by HU. Additionally, a blocking lesion on the leading strand can generate a long stretch of ssDNA on which repriming processes produce multiple 5′-ended junctions, thus contributing to the amplification of ATR signaling. Furthermore, in the case of a fork stalling lesion where ssDNA exposure is not prevalent, such as at ICLs, fork reversal may be involved to generate an appropriate 5′-ended junction by DNA remodeling, where additional DNA nucleases (e.g., DNA2) or translocases (e.g., FANCM) may participate in generating a checkpoint-competent structure [71,112,113]. In contrast to TOPBP1, ETAA1 is recruited to ssDNA via its direct interaction with RPA, suggesting that the extent to which ETAA1 triggers ATR activation may be proportional to the levels of persistent ssDNA accumulation [114,115]. The pathway of ATR activation by either TOPBP1 or ETAA1 may produce distinct outputs for checkpoint signaling depending on DNA lesion structures and the extent of DNA damage. Indeed, while TOPBP1 is necessary for ATR activation in response to replication stress, ETAA1 was shown to specifically engage ATR activation to control S-G2 cell cycle transition and mitosis in an unchallenged condition [116,117].

Once activated, one of the major downstream targets of ATR to propagate the replication checkpoint signaling is CHK1. While CHK1 is a relatively specific target of ATR under replication stress, ATR-independent phosphorylation by ATM or DNA-PK has also been reported [118,119]. CHK1 phosphorylation occurs at multiple sites, and several studies support that ATR-dependent sequential phosphorylation events at Ser317 and Ser345 are necessary for maximal checkpoint activation [120,121]. The Ser345 phosphorylation may help relieve the N-terminal kinase domain from the intramolecular autoinhibition mediated by the C-terminal regulatory region [122]. Furthermore, autophosphorylation of CHK1 on Ser296 also occurs that is contingent on the prior phosphorylation of Ser317 and Ser345, which is necessary for the spread of CHK1 signals [120,121].

One major regulatory mechanism underlying the canonical activation of CHK1 via ATR is the interaction between the FPC and the RPA-coated ssDNA. TIPIN within the FPC directly binds to the RPA2 subunit of the RPA complex, which in turn recruits CLASPIN to stalled forks [37,123]. Depletion of either CLASPIN or TIPIN, or expression of a TIPIN mutant that cannot interact with RPA, impairs CHK1 phosphorylation in response to a variety of replication stressors in vitro, in *Xenopus* egg extracts, and in human cells, supporting the existence of the RPA-TIPIN-CLASPIN axis for the activation of CHK1 [124,125,126,127,128]. The direct interaction between CLASPIN and CHK1 indicates that an ssDNA-exposed stalled fork acts as a platform that brings the replisome, ATR, and CHK1 in close proximity to facilitate checkpoint activation. Indeed, a recent structural study revealed that the CLASPIN-CHK1 interaction does not affect the catalytic activity of CHK1, implicating a role of CLASPIN as a scaffold for CHK1 activation by ATR [129]. AND-1, another component of the FPC, also promotes the association of CLASPIN to ssDNA, and thus the CLASPIN-CHK1 interaction, thereby potentiating ATR-dependent CHK1 activation [130].

### 4.2. Functions of the ATR Pathway

#### 4.2.1. Cell Cycle Checkpoint

One prominent role of the ATR checkpoint is to activate cell cycle arrest under replication stress to delay cell cycle progression and allow time for DNA repair (Figure 4A). When CHK1 is activated, it triggers cell cycle arrest mainly by phosphorylation and inactivation of the CDC25 phosphatases. Phosphorylation of CDC25A targets it for proteasomal degradation, which antagonizes its phosphatase activity to remove the inhibitory phosphorylation of CDK2 and CDK1, and thus suppresses the action of Cyclin A-CDK2 and Cyclin B-CDK1 to trigger intra-S and G2/M checkpoints, respectively [131,132]. In this process, a series of phosphorylation events on CDC25A recruits β-TrCP ubiquitin E3 ligase to polyubiquitinate and degrade CDC25A [133,134]. In contrast, CDC25C phosphorylation by CHK1 creates a docking site for 14-3-3, which sequesters CDC25C to the cytosol, thus inactivating Cyclin B-CDK1 and preventing mitotic entry [135,136]. CDC25B is mainly localized in the cytosol, but its phosphorylation by the centrosome-associated CHK1 negatively regulates its activity at the centrosome to inhibit CDK1 and prevent premature initiation of mitosis [137,138].

#### 4.2.2. Control of Replication Origin Firing

Only a handful of licensed origins are selected to fire, while the majority of origins remain dormant and are passively replicated in the S phase. It is estimated that about 50,000 origins are chosen from an excess of 500,000 licensed origins in the human genome in a stochastic manner [139]. While these dormant origins are considered inactive during S phase progression, the surplus of MCMs in the pre-RC complex is necessary for restraining fork speed and maintaining the symmetry of bidirectional replication forks [140]. By contrast, under replication damage, local dormant origins in the vicinity of stalled forks fire as backup origins to complete DNA synthesis, whereas origin firing in late-replicating regions is inhibited to avoid additional replication stress and depletion of dNTPs. The ATR kinase plays an essential role in controlling the firing of replication origins both during unchallenged fork progression and in the event of DNA damage associated with replication stress (Figure 4B). Original studies in *Xenopus* egg extracts revealed that ATR inhibition with caffeine or neutralizing antibodies triggers excessive origin firing in the early S phase, implicating a basal-level checkpoint that suppresses unnecessary origin firing [141,142]. CHK1 deficiency and subsequent premature activation of Cyclin A2-CDK1 leads to late origin firing in the early S phase [143]. Compromised CDK2 activity during the S phase due to the inactivation of ATR, CHK1, or WEE1 kinases results in a significant increase in origin firing and reduction in fork speed [118,144,145]. Furthermore, ATR was shown to suppress CDK1-dependent FOXM1 phosphorylation in the S phase, enforcing the S/G2 checkpoint to prevent a premature S/G2 transition [116]. A recent study also revealed that suppression of CDK1 activity by ATR-CHK1 stabilizes the interaction between replication timing regulatory factor 1 (RIF1) and phosphatase 1 (PP1), allowing cells to dephosphorylate CDC7 and CDK2 substrates, and thus inhibit CMG assembly and activation [146].

Multiple pathways exist to control origin firing by ATR and CHK1. ATR-CHK1 suppresses CDK and DDK activities, thereby impairing the phosphorylation of TRESLIN (a functional homolog of Sld3 in *S. cerevisiae*) [147] and MCMs [148], respectively, and subsequently blocking the assembly of CDC45 to the pre-RC complex [149,150,151]. Additionally, phosphorylation of mixed lineage leukemia (MLL) by ATR under genotoxic stress stabilizes its intracellular levels, which promotes histone H3K4 methylation at late replication origins and inhibits CDC45 loading, thus enforcing the intra-S checkpoint [152]. Accordingly, the t(11;16) MLL fusion protein present in leukemia functions as a dominant negative that abrogates the replication checkpoint. Blocking CDC45 loading by damage-induced phosphorylation of Sld3 by Rad53 or TRESLIN by CHK1 inhibits late origin firing, indicating that the phosphorylation of multiple replication factors is an important mechanism of negative regulation to suppress origin firing [147,153].

On the other hand, a local origin within existing replication factories preferentially needs to be activated when replication fork progression is impaired [154]. Suppression of ATR or CHK1 activity is associated with a decrease in the rate of fork progression and density of active replication origins, as well as an increase in under-replicated regions of the genome [145,155,156,157]. At least one mechanism involves phosphorylation of the MCM helicase complex, for instance ATR-dependent phosphorylation of MCM2 on S108 in response to replication damage in mammalian cells [158]. In *Xenopus* egg extracts, ATR-dependent phosphorylation of MCM2 on S92 promotes the recruitment of Plx1, the ortholog of Plk1, to inhibit CHK1 activity, thus allowing for the association of CDC45 to a nearby dormant origin and derepression of DNA replication initiation [159]. FANCI, a key component of the FA DNA ICL repair pathway, was shown to be another important target of ATR in controlling local and global origin firings [160]. While FANCI is required for dormant origin firing by CDC7-dependent activation of MCMs, ATR-mediated FANCI phosphorylation in response to an increase in replication stress inhibits its activity while switching its role to promote replication fork restart and DNA repair. The role of the ATR-CHK1 pathway in promoting replication under DNA damage may extend beyond the control of local origin firing; CHK1 inactivates APC/C^Ddh1^ ubiquitin E3 ligase to stabilize CDC7-DBF4, which promotes chromatin binding of RAD18 required for DNA lesion bypass, suggesting that replication origin control may be orchestrated with DNA repair and recombination processes [161].

#### 4.2.3. Protection and Recovery of Stalled Replication Forks

While inducing cell cycle arrest and limiting the number of available replication origins is a critical function of the replication checkpoint, ATR also works locally at stalled forks to protect the damaged fork and facilitate its recovery to continue replication. Prolonged fork stalling is expected to accumulate ssDNA gaps and breaks that ultimately lead to irreversible fork collapse, i.e., the condition in which a replication fork is not able to support DNA synthesis. While the disassembly of the replisome as a prerequisite for fork collapse is being debated [162,163], the ATR-CHK1 pathway is known to directly modulate nuclease activities and fork remodeling to stabilize stalled forks (Figure 4C).

Earlier studies in *S. cerevisiae* demonstrated that the replication checkpoint mediated by Mec1 and Rad53 is responsible for stabilizing stalled replication forks and preventing damage-induced fork breakage [164,165]. ATR and its downstream CHK1 are also essential for preventing fork breakage during unchallenged fork progression and replication of fragile sites in mammalian cells [166,167]. ATR inhibition leads to significant fork collapse and cell lethality in human cells, which is dependent on SLX4-dependent fork breakage that yields DSBs and CtIP-dependent resection that generates excess ssDNA at both template and nascent DNA strands. It was shown that ATR phosphorylates SMARCAL1 on S652, which restricts its fork regression and processing activities, suggesting that ATR negatively regulates fork reversal or other DNA remodeling steps such as Holliday junction formation to prevent aberrant fork processing that would lead to fork collapse [168]. Consistently, fork reversal is frequently observed in Rad53 replication checkpoint mutants in *S. cerevisiae*, and Exo1 nuclease is responsible for counteracting reversed fork accumulation [164,169,170]. Accordingly, a loss of balanced intracellular SMARCAL1 levels causes replication fork collapse in an SLX4-dependent manner [171]. The SUMO-targeted ubiquitin E3 ligase RNF4 and activation of the Aurora kinase A (AURKA)-PLK1 pathway also involves excessive fork processing that is dependent on SLX4 in the absence of ATR activity [172]. Notably, CDK regulates structure-specific nucleases including SLX1-SLX4 and MUS81-EME1 at the G2/M transition to resolve under-replicated foci and Holliday junctions [173]. Therefore, aberrant CDK activity may contribute to the premature activation of nucleases and processing of stalled forks [174]. In checkpoint-deficient settings such as WEE1 or CHK1 inhibition and upregulated CDK activity, aberrant fork processing and DSB formation have been attributed in part to MUS81 activity [144,175,176,177]. Similarly, premature phosphorylation of SLX4 by CDK1 was shown to promote the recruitment of MUS81 at replication forks, causing DSBs and fork collapse [178]. While MUS81 is necessary for the formation of DSBs in response to replication stress, loss of MUS81 also prevents the timely recovery of stalled forks and increases chromosome aberration, suggesting that the activity of MUS81 should be tightly controlled for the proper processing of a stalled fork into an intermediate favorable to DNA repair [179]. Furthermore, inhibition of the MRE11 nuclease activity not only suppresses ssDNA accumulation but also MUS81-dependent DSB formation upon CHK1 inhibition, indicating that MRE11 provides a link between CDK activation and the unscheduled structure-specific nuclease activity from MUS81 [180,181].

It is generally accepted that replication problems lead to DNA damage, as shown in CHK1- or WEE1-deficient cells, which exhibit slow fork movement and increased DNA breaks caused by CDK hyperactivation and increased origin density. However, a study demonstrated that inhibition of MUS81-EME2 and MRE11 nuclease activities is sufficient to restore normal fork progression and a density of initiation events in CHK1-deficient cells, suggesting that nuclease-dependent DNA damage events elicited by the loss of CHK1 activity can act upstream of changes in replication dynamics [180]. dNTP pools were shown to be rate-limiting in this condition, suggesting that cells may adapt to DNA damage by redistributing dNTPs to repair and fork stabilization, thereby limiting precursors for replication and slowing down fork progression in response to aberrant fork cleavage and resection.

Proper protection of stalled forks allows for timely replication fork restart, and the ATR-CHK1 pathway is also known to be involved in facilitating stalled fork recovery. In response to Pol α inhibition that rapidly reduces DNA synthesis, increasing CHK1 activity by ectopic CHK1 expression leads to the phosphorylation of PRIMPOL that is necessary for promoting repriming on the leading strand template [182]. Intriguingly, in comparison to HU-induced damage, CHK1 activity is less pronounced while ATR becomes active upon Pol α inhibition, in which case fork reversal is favored, suggesting that distinct ATR and CHK1 activities may modulate the pathways of replication stress tolerance. ATR also phosphorylates some of the TLS polymerases including REV1 and Pol η, indicating that ATR is directly involved in lesion bypass [183,184]. Additionally, the WRN and BLM helicases are direct phosphorylation targets of ATR [185,186]. Notably, the ATR signal can spread out to mediate global fork reversal at replication forks that are not directly challenged by a DNA ICL lesion, indicating that global remodeling of ongoing forks may be an important mechanism for ATR to restrict fork progression and prevent breakages [187].

#### 4.2.4. Restriction of the Replisome Activity

The ATR-CHK1 pathway directly modifies replisome components to restrain DNA replication and stabilize stalled forks. One target of its negative regulation is the FPC, presumably attenuating its replication-promoting function under DNA replication damage (Figure 4D). In *S. cerevisiae*, Rad53-deficiency under HU damage results in uncoupling of leading and lagging strand synthesis, in which DNA synthesis progresses preferably along the lagging strand, thereby exposing long stretches of single-stranded leading strand templates [188]. Surprisingly, this asymmetric DNA synthesis was also observed in an Mrc1 replication checkpoint-deficient mutant that is not able to be phosphorylated by Rad53, but not in an Mrc1-knockout mutant in which both its replication and checkpoint functions are missing, indicating that Mrc1 activity becomes unrestrained if it fails to be phosphorylated [189]. Deletion of Tof1 in the Rad53-mutant also suppressed asymmetric DNA synthesis under replication stress, suggesting that one important function of the replication checkpoint is to restrict the replisome activity to prevent excessive replisome uncoupling at stalled forks. Similarly, it was shown that Mrc1 phosphorylation by Rad53 slows down replication fork elongation in vitro [190]. Given that CLASPIN/Mrc1 is a major element in the FPC that stimulates fork progression in vitro, modification of CLASPIN/Mrc1 by the replication checkpoint may be necessary to restrict fork elongation thus prevents replisome uncoupling. Notably, pharmacological inactivation of Pol α is sufficient to uncouple leading and lagging strands, thereby generating an excessive amount of ssDNA in a level higher than ATR inhibition and HU treatment, further supporting the notion that strand coupling needs to be tightly kept in check to prevent fork collapse [191].

ATR is known to phosphorylate FANCD2, a key element for FA pathway activation, which promotes its association to MCM2-7 at nascent DNA in response to replication arrest [192]. This is necessary for restraining DNA synthesis and minimizing MRE11-dependent fork resection, which operates independently of DNA ICL repair. FANCD2 phosphorylation correlates with MCM2-7 phosphorylation by ATR, suggesting that ATR may collectively modify the replisome and its regulators to control fork progression and restrict nuclease activity.

### 4.3. Checkpoint Failure and Replication Catastrophe

Emerging evidence supports the notion that the amount of ssDNA generated at stalled forks is a key determinant of whether a replication fork will break down, leading to irreversible fork collapse. All forms of ssDNA, regardless of its replication stress origin, demand protection from RPA, which binds stretches of ssDNA and acts as a buffer to prevent their exposure beyond the threshold of protection (Figure 5). Consequently, depletion of the available nuclear pool of RPA due to excessive ssDNA accumulation, termed RPA exhaustion, leads to genome-wide breakage and collapse of replication forks, a phenomenon referred to as replication catastrophe [193]. Pharmacological or genetic inactivation of the replication checkpoint proteins, including ATR, CHK1, and WEE1, is sufficient to accumulate ssDNA and trigger fork breakage in the absence of exogenous genotoxic stress, suggesting that the ATR pathway is essential for suppressing ssDNA production and ensuring normal fork progression [118,144,167,175,176,180]. A recent high-resolution imaging and computational modeling study revealed that the basal-level activity of ATR constantly monitors and regulates the amount of RPA at active forks independently of CHK1, and increased ATR-RPA contacts in response to exposed ssDNAs amplifies the activity of ATR to engage the canonical ATR-CHK1 checkpoint [194]. This suggests that the role of ATR in surveying the RPA content near the replication machinery is critical for avoiding replication catastrophe. The global restriction of origin firing by ATR is likely to ensure that a surplus amount of RPA is available for providing local protection of active forks from breakage, suggesting that ATR integrates both local and global genome surveillance mechanisms to limit ssDNA exposure. In line with this idea, our recent study implementing an auxin-inducible degron to rapidly degrade TIM at ongoing forks revealed the concerted roles of ATR that operate both locally and globally to prevent replication catastrophe against the acute replisome dysfunction triggered by TIM loss as a model for endogenous replication stress [195].

The model describing RPA exhaustion and ensuing replication catastrophe predicts that infliction of DNA replication stress over the RPA protection threshold either by cytotoxic chemotherapy or replication checkpoint inhibitors underlies the clinical response of cancer cells to the high levels of replication stress [196]. In this sense, replication inhibitors or DNA damaging agents can be combined with checkpoint inhibition to accelerate the accumulation of ssDNA, in addition to triggering excessive origin firing, depending on the extent of endogenous DNA replication stress of cancer cells. Accordingly, hallmarks of replication catastrophe have been proposed as a biomarker, for instance, to predict the sensitivity to poly(ADP-ribose) glycohydrolase (PARG) inhibitors in subsets of ovarian cancer [197,198]. The model also predicts that irreversible fork collapse is a consequence of fork breakage in the absence of checkpoint function. The genome-wide detection of ssDNA and mapping of DNA breaks after HU exposure in *S. cerevisiae* revealed that ssDNA is detected prior to chromosome breakage, indicating that ssDNA is a precursor of collapsed forks [199]. While it is largely anticipated that ssDNA is accumulated ahead of stalled replicative polymerases due the replisome uncoupling, remodeling of stalled forks and degradation of nascent DNA strands could also be a source of extensive ssDNA generation. Notably, the synergistic fork instability manifested in acute TIM degradation and ATR inhibition was dependent on the aberrant nucleolytic degradation of ssDNA-exposed stalled forks, indicating that replication catastrophe is an active process that involves the dynamic, often aberrant, remodeling and processing of replication forks [195].

## 5. Checkpoint Inhibitors for Cancer Therapy

Triggering replication catastrophe by exacerbating DNA replication stress has become one of the key mechanisms to target cancer cells with chronic replication problems and cell cycle deregulation [200]. Many preclinical studies have highlighted the antitumor effects of checkpoint inhibitors as mono- or combination therapies with cytotoxic agents, and based on these studies, checkpoint inhibitors are currently being evaluated as cancer therapeutics in multiple clinical trials. Here, we review the current development of checkpoint inhibitors that target ATR, CHK1, and WEE1 kinases.

### 5.1. ATR Kinase Inhibitors

The most notable ATR inhibitors (ATRi) include VE822, ceralasertib (AZD6738), M4344, and BAY-1895344. VE822 (M6620/VX-970, a VE821 analogue) was one of the first ATRi to move into clinical trials following data showing anti-tumor effects in preclinical in vitro and in vivo patient-derived xenograft (PDX) models [201]. Recent Phase I trials showed that VE822 is well-tolerated and effective when combined with veliparib (PARP inhibitor; PARPi), cisplatin, or topotecan; there are currently 14 ongoing Phase I/II trials investigating VE822 efficacy in combination with gemcitabine, platinum-based agents, and topoisomerase poisons. Another highly selective, orally bioavailable ATRi, ceralasertib (AZD6738, an AZ20 analogue), has been shown to have single-agent efficacy in ATM-deficient and p53-deficient cancer cell lines and tumor models, with synergistic anti-proliferative effects when combined with chemotherapeutic and DNA damaging agents [202,203,204]. Three completed Phase I/II clinical trials recently evaluated ceralasertib as a single agent in relapsed/refractory ATM-deficient chronic lymphocytic leukemia (CLL), in combination with paclitaxel for non-responsive metastatic cancer, and in combination with olaparib for relapsed small cell lung cancer (SCLC). To date, there are 28 ongoing trials utilizing ceralasertib as a single agent or in combination with gemcitabine, ionizing radiation (IR), PARPi, immunotherapy (trastuzumab) and other DNA damaging agents (i.e., cisplatin, etoposide, and taxanes). M4344 (VX-803), an orally bioavailable ATP-competitive ATRi, previously shown to have synergistic effects when combined with topoisomerase poisons in patient-derived tumor organoid and xenograft models, recently underwent a dose escalation study in combination with carboplatin [205]. Interestingly, BAY-1895344, a highly potent and selective ATRi with strong anti-proliferative effects in various cancer cell lines and xenograft models, was recently studied in a human Phase I trial, reporting that it was well tolerated and exhibited anti-tumor activity in refractory solid tumors, and non-Hodgkin’s and mantle cell lymphomas with ATM deficiencies [206,207]. A single-agent expansion phase study is currently being conducted in DNA damage response (DDR)-deficient cancers and additional combination studies with pembrolizumab and niraparib, a PARPi.

### 5.2. CHK1 Kinase Inhibitors

It can be implied that the anti-cancer effects of CHK1 inhibitors (CHK1i) may not differ much from ATRi, as they are part of the same signaling pathway. However, various studies revealing CHK1 overexpression or ATR-independent CHK1 activation in cancer make CHK1 an independent target for cancer therapy. UCN-01, a first-generation multi-target serine-threonine protein kinase inhibitor that also targets CHK1, sensitized p53-deficient cancer cells to DNA damaging agents [208,209,210,211]. Although it was not considerably potent, several Phase I/II clinical trials utilizing UCN-01 in combination therapies further highlighted the therapeutic potential of targeting CHK1. Subsequential development of CHK1i, such as SRA737 (formerly CCT244747) and prexasertib (LY2606368), have shown great promise in preclinical and clinical models. SRA737, a novel, potent and orally bioavailable CHK1i, has been shown to have single-agent efficacy in MYC-driven cancers, and in combination with IR, gemcitabine, irinotecan, and PARPi [212,213]. Recent Phase I/II trial data support further development of SRA737 in combination therapies [214,215]. Finally, prexasertib, a potent CHK1i, has been shown to induce antitumor effects through the induction of replication stress, checkpoint abrogation, DSB formation, and ultimately replication catastrophe [216]. Additional studies have shown synergistic effects of prexasertib when combined with olaparib in TNBC, ovarian, and gastric cancers and additionally with cisplatin in platinum-refractory cancers [217,218,219,220]. There are currently 19 completed and ongoing Phase I/II trials that are evaluating prexasertib as a mono- or combination therapy in p53-deficient, DDR-deficient, or CCNE1-amplified cancers.

### 5.3. WEE1 Kinase Inhibitors

To date, there are a handful of WEE1 inhibitors that have been studied in various preclinical models and are currently in Phase I/II trials, which include AZD1775 (adavosertib), Zn-c3, IMP7068, Debio 0123, and SY-4835. One of the most well-characterized is AZD1775, a potent and highly selective WEE1i that has shown anti-tumor effects, independent of p53 mutational status, in various in vitro and in vivo cancer models as a single-agent or in combination with chemotherapy or IR [221,222]. More specifically, AZD1775 sensitizes cancer cells to therapies that interfere with DNA synthesis and repair processes (i.e., HR), suggesting that certain DNA replication/repair proteins can serve as key determinants of AZD1775 sensitivity, in addition to p53 status [223]. Reported Phase I/II trial data further suggest that AZD1775 is highly tolerated and exhibits anti-tumor effects primarily by potentiating the cytotoxic effects of chemotherapeutic agents used in combination [224]. Currently, 18 ongoing Phase I/II trials are assessing AZD1775 as a mono- or combination therapy with antimetabolites (gemcitabine or cytarabine), taxanes, platinum agents, irinotecan, and PARPi in various solid tumors. Recently, ZN-c3, a novel orally bioavailable small molecule inhibitor, was shown to have a higher WEE1 selectivity and safety profile compared to AZD1775 with similar levels of efficacy in in vitro and in vivo [225]. ZN-c3 quickly transitioned into Phase I/II trials to assess its effects as a mono- or combination therapy with niraparib, gemcitabine, encorafenib, carboplatin, doxorubicin, or immunotherapies. There are still a multitude of ongoing trials with additional WEE1 inhibitors (IMP7068, Debio 0123, and SY-4835), further highlighting the exciting potential of WEE1i in the development of novel cancer therapeutics.

## 6. Conclusions

This review has highlighted the various adaptive responses of a stalled fork to tolerate DNA damage and efficiently resume DNA replication. These processes involve a dynamic remodeling of stalled forks supported by specialized polymerases, nucleases, and helicases, whose unique activities determine the best course of DNA transactions for rapid fork protection and recovery. The ATR-CHK1 checkpoint acts as a master regulator of replication fork integrity, which not only directly affects fork stability but also controls global origin activation and cell cycle progression. In the absence of these key regulators, extensive ssDNA accumulation and aberrant fork processing/degradation ensues, which drives tumorigenesis that is often associated with genetic defects in DNA replication and repair as well as stalled fork protection. As evidenced by multiple ongoing efforts for checkpoint inhibitor development, deregulation of fork plasticity represents a unique replication vulnerability of cancer cells. We envision that the replisome itself is a hub that senses replication damage, engages the checkpoint response, and coordinates the activities of DNA-protein transactions necessary for overcoming DNA damage. Future studies to elucidate the fate of the replisome during fork remodeling and different layers of posttranslational modifications that allow for the plasticity of a stalled fork will certainly help better understand the complexity of responses to replication damage, providing a new route to refined cancer therapy and the discovery of replication stress-specific biomarkers.

## Figures and Tables

**Figure 1 ijms-24-10488-f001:**
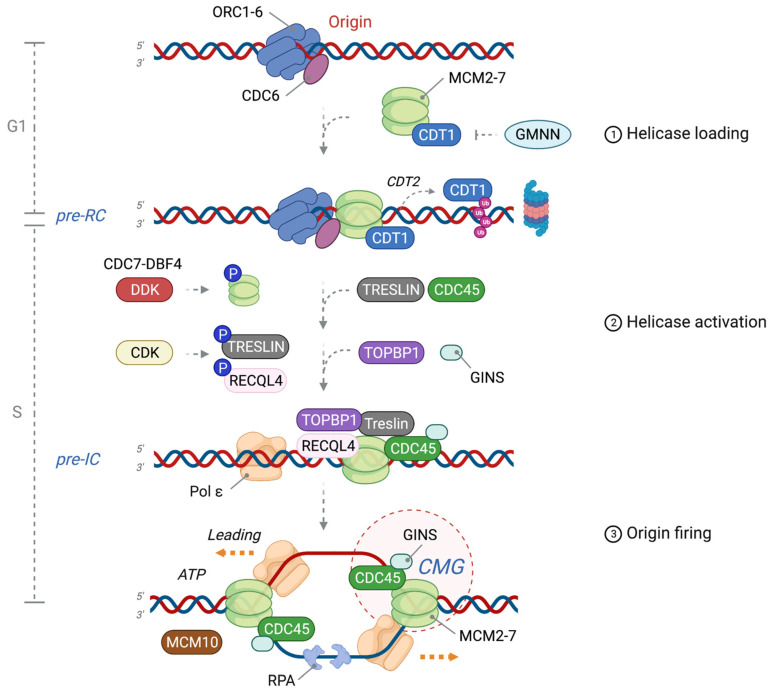
Licensing and activation of a replication origin. The replication origin of eukaryotes is first marked by the origin recognition complex 1-6 (ORC1-6), which recruits cell division cycle 6 (CDC6) and subsequently loads minichromosome maintenance 2-7 (MCM2-7)/Cdc10-dependent transcript 1 (CDT1) to form the pre-replication complex (pre-RC); one of two double hexamers is shown. An excess of MCM2-7 hexamers is loaded into DNA during the G1 phase, constituting dormant replication origins. Licensing occurs once per cell cycle and is restricted to G1 by the inhibitory interaction of CDT1 with GEMININ (GMNN) during S and G2 phases (in vertebrates), as well as proteasomal degradation of CDT1. The pre-initiation complex (pre-IC) is formed by DBF4-dependent kinase (DDK)-mediated phosphorylation of MCM2-7, which promotes binding of TRESLIN and CDC45 to replication origins. CDKs subsequently phosphorylate TRESLIN and RECQL4, thereby promoting the recruitment of TOPBP1 and Go-Ichi-Ni-San (GINS1-4) complex to assemble the CMG (CDC45-MCM2-7-GINS) helicase complex together with polymerase ε (Pol ε). Origin activation: engagement of pre-IC with MCM10 initiates melting of dsDNA and unwinding by the CMG. RPA binds and protects resulting ssDNA, and two separate replisomes are established to synthesize nascent DNA in a 5′-3′ direction. ℗: phosphorylation.

**Figure 2 ijms-24-10488-f002:**
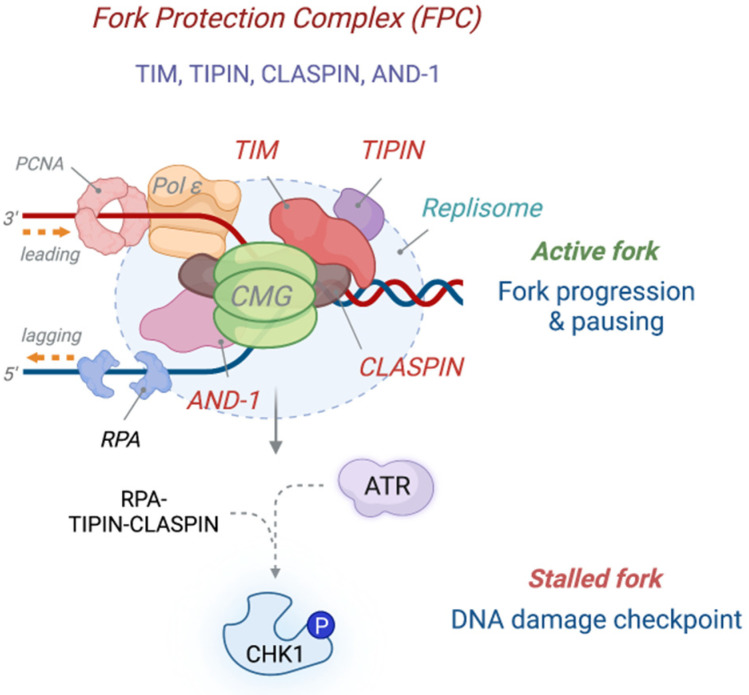
The structure and function of the fork protection complex. The fork protection complex (FPC) is part of the replisome, consisting of the TIMELESS (TIM)-TIPIN heterodimer, CLASPIN, and AND-1. It acts as a scaffold to couple the CMG helicase and replicative polymerase activities to limit ssDNA exposure and ensure unperturbed fork progression. CLASPIN promotes replication fork progression, whose activity is augmented by TIM-TIPIN. The positioning of TIM-TIPIN at the front of the CMG helicase not only allows the FPC to grip the DNA duplex and stabilize the replisome at ongoing replication forks, but also restricts the CMG helicase activity to suppress aberrant uncoupling of replisome activity, which acts as a pausing mechanism to restrain DNA replication under replication damage or navigate through the difficult-to-replicate genome. In response to DNA damage, direct interaction of TIPIN with RPA brings CLASPIN and CHK1 into proximity with ataxia telangiectasia and Rad3-related (ATR) at stalled replication forks, in turn facilitating ATR-dependent CHK1 phosphorylation to initiate the DNA damage checkpoint. Thus, the FPC is a key regulatory element of the replisome to control DNA replication and checkpoint responses at both active and stalled replication forks. ℗: phosphorylation.

**Figure 3 ijms-24-10488-f003:**
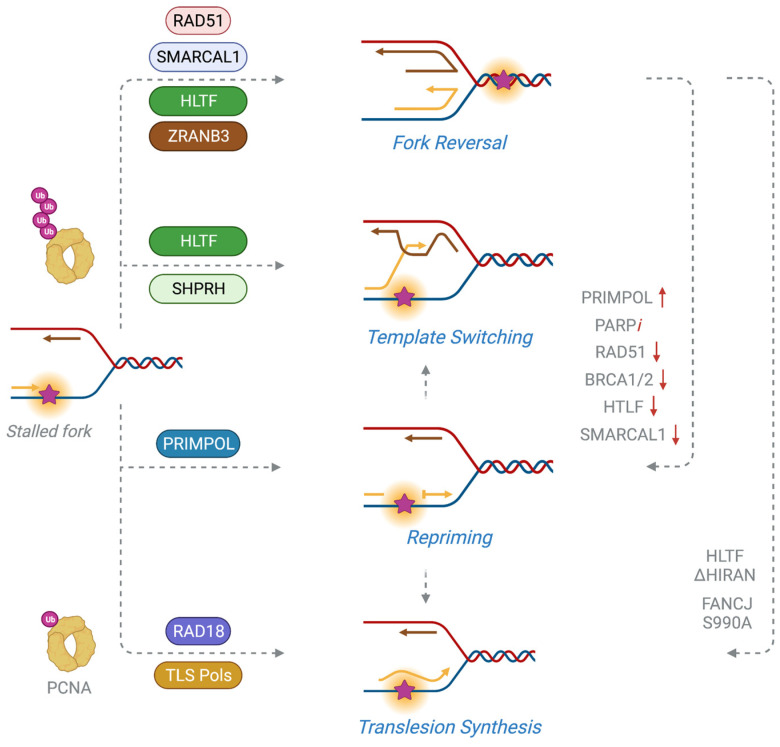
Responses of a stalled fork to DNA replication damage. A stalled DNA replication fork engages distinct DNA damage tolerance (DDT) pathways to promote fork recovery and timely fork restart. Fork reversal by the coordinated action of the RAD51 recombinase and fork remodelers (i.e., SMARCAL1, ZRANB3, and HLTF) generates a four-way chicken foot structure that stabilizes a DNA lesion and facilitates fork restart. PCNA polyubiquitination catalyzed by HLTF and SHPRH engages the template switching (TS) mechanism in an error-free manner, which occurs between two nascent strands within the same replication fork of sister chromatid pairing. DNA replication can also be restarted beyond a DNA lesion via repriming that is dependent on DNA-directed primase/polymerase PRIMPOL. Repriming leaves daughter-strand ssDNA gaps that often need to be repaired post-replicatively. Translesion DNA synthesis (TLS) is activated by RAD18-dependent PCNA monoubiquitination, which recruits specialized TLS polymerases to help on-the-fly bypass a DNA lesion on the leading strand. Due to the error-prone nature of TLS polymerases, TLS is highly mutagenic. ssDNA gaps formed by repriming can be filled either by TS or TLS that acts in post-replicative repair outside of the S phase. Balancing between fork reversal, repriming, and TLS is determined by types of the DNA lesion, extent of fork stalling, certain loss-of-function genetic backgrounds, and PCNA modification. ↑: upregulation, ↓: downregulation, *i*: inhibition.

**Figure 4 ijms-24-10488-f004:**
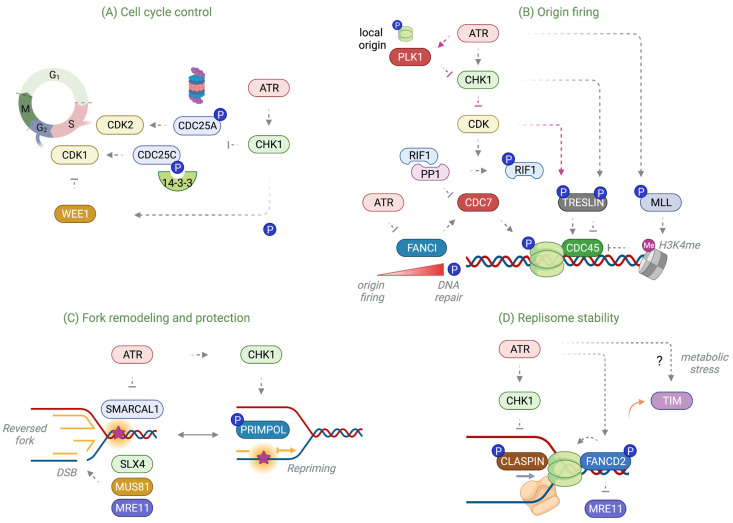
Checkpoint pathways regulated by ATR. (**A**) Phosphorylation of CHK1 by ATR is critical for triggering cell cycle arrest under DNA damage. When activated, CHK1 phosphorylates and inactivates the CDC25 phosphatases, CDC25A, CDC25B, and CDC25C, either by rapid proteasomal degradation or sequestration into the cytoplasm by 14-3-3, which in turn elevates the inhibitory phosphorylation of CDK1 or CDK2 produced by the CHK1-activated WEE1 kinase. (**B**) ATR controls dormant origin firing. Active CHK1 negatively regulates CDK-mediated phosphorylations at its origins, blocking CDC7-mediated CDC45 loading. One mechanism involves suppression of RIF1 phosphorylation by CDK, thereby keeping a stable RIF1-PP1 phosphatase complex to inhibit CDC7. CHK1 also directly phosphorylates TRESLIN, and thus limits CDC45 loading to its origins. An additional regulatory factor of CDC7 includes FANCI, which promotes dormant origin firing, yet is counteracted by phosphorylation by ATR, thereby switching its role in DNA repair under increasing levels of replication stress. Additionally, phosphorylation and stabilization of MLL by ATR promotes MLL association with chromatin, which catalyzes histone H3 methylation at lysine 4 (H3K4me) and blocks CDC45 to its origins. In contrast, inhibition of CHK1 activity in the vicinity of a stalled fork by MCM2 phosphorylation and PLK1 recruitment by ATR allows for CDC45 loading and activation of local origins (magenta arrows). (**C**) ATR restricts unnecessary fork reversal driven by SMARCAL1, which is otherwise subject to aberrant fork processing by nucleases. The ATR-CHK1 pathway may directly govern pathway choices between fork reversal and repriming by promoting PRIMPOL activity upon fork stalling. (**D**) Phosphorylation of CLASPIN by ATR-CHK1 constrains the replication-promoting function of CLASPIN to allow a replication fork to pause in response to replication damage. ATR-dependent FANCD2 phosphorylation promotes its association to the CMG, which restricts fork progression and keeps MRE11 activity in check. Dissociation of TIM from the replisome under metabolic stress occurs to slow down fork progression (orange arrow), but whether ATR regulates this stress response remains unclear (question mark).

**Figure 5 ijms-24-10488-f005:**
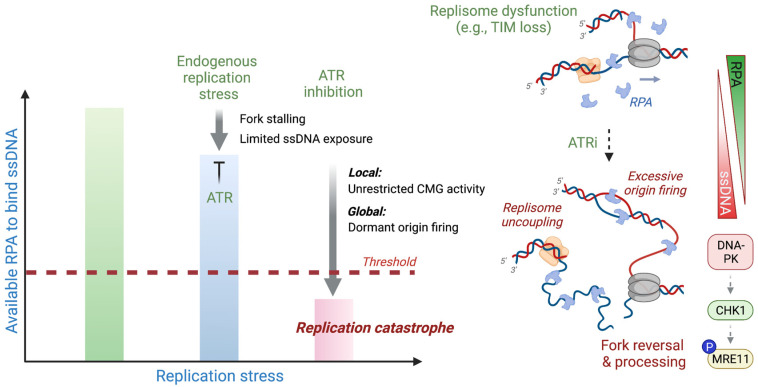
Role of ATR in counteracting replication catastrophe ssDNA requires protection by RPA to prevent DNA breakage. A great excess of RPA is available in cells to act as a buffer to suppress replication stress or DNA damage. Endogenous replication stress caused by stochastic replisome uncoupling or dysfunction (e.g., rapid TIM loss at active forks) exposes limited ssDNA and stalls replication forks, which engages the ATR replication checkpoint to prevent excessive accumulation of ssDNA. ATR integrates both local fork protection, i.e., restricting the CMG activity, and global fork protection, i.e., inhibiting dormant origin firing. Accordingly, hyper-reliance on ATR renders cells sensitive to ATR inhibition by exacerbating the amount of ssDNA, leading to RPA exhaustion below the threshold of protection, and thus replication catastrophe. This synergistic fork instability involves reversal and aberrant processing of a stalled fork with extensive ssDNA, in part mediated by DNA-PK-CHK1-dependent MRE11 activity [195].

## Data Availability

Not applicable.

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
