# Peer review of "The Adaptive Mechanisms and Checkpoint Responses to a Stressed DNA Replication Fork"

_ijms, 2023, doi:10.3390/ijms241310488_

Round 1

Reviewer 1 Report

Saldanha et al., reviewed the different cellular responses activated by stalled replication fork when DNA damage occurs. The authors nicely revised the key players involved in stalled replication fork remodeling as well as the activities exerted for rapid fork protection and recovery. The authors mainly focused on the ATR-CHK1 checkpoint since it acts as a master regulator of replication fork integrity and at the end, briefly examined the use of replication stress specific markers in cancer therapy.

The Review is very interesting, exhaustive, well written and organized. Figures are greatly done, clear and informative. Literature is fully cited.

However, I would suggest to adjust the Abstract because does not summarize properly the content of the Review, and I would add few references to the intro of section 5, “Checkpoint inhibitors for cancer therapy”.

Author Response

Saldanha et al., reviewed the different cellular responses activated by stalled replication fork when DNA damage occurs. The authors nicely revised the key players involved in stalled replication fork remodeling as well as the activities exerted for rapid fork protection and recovery. The authors mainly focused on the ATR-CHK1 checkpoint since it acts as a master regulator of replication fork integrity and at the end, briefly examined the use of replication stress specific markers in cancer therapy.

The Review is very interesting, exhaustive, well written and organized. Figures are greatly done, clear and informative. Literature is fully cited.

However, I would suggest to adjust the Abstract because does not summarize properly the content of the Review, and I would add few references to the intro of section 5, “Checkpoint inhibitors for cancer therapy”.

We have edited Abstract to better represent the content of the manuscript. We have included a reference that provides an overview of checkpoint inhibitors and cancer therapy in the intro of section 5.

Reviewer 2 Report

This review article focuses on how ATR-dependent checkpoint pathways are coordinated with the mechanism of DNA replication. In this content, author included details of the regulation of DNA replication (initiation and the roles of many components in the replisome) as well as the response of replication forks to various replication stress as well as DNA damage. I think this in-depth content will update reader’s understanding of the mechanisms of DNA replication and its ‘adaptive mechanisms’.

I listed some specific points which is unclear or can be improved (listed below).

-       The second paragraph of ‘1, initiation of DNA replication’ (line 40-55): this part is based on the research in budding yeast. I suggest authors to mention this point.

-       Recent paper also mentioned redundant roles of CDC7 and CDK1 during G1/S transition in mammalian cells (https://doi.org/10.1038/s41586-022-04698-x). This research could be incorporated in this part (or somewhere relevant).

-       Line 103 & 115: I suggest to clarify what the pre-initiation complex (or what proteins consist of this complex). Reference: https://doi.org/10.15252/embr.201744206  

-       Line 124 ‘replication fork stalling results in a large stretch of ssDNA accumulation’: I do not think fork stalling itself causes a stretch of ssDNA.

-       Line 142-146 ‘This unique configuration ~ MCM central pore’: These sentences are unclear to me. Is it possible to clarify this part with figure 2?

-       Line 206-208 ‘Perhaps ~ TIM may ~ suppress R-loop~’. are there any evidences? If not, I suggest authors to remove this sentence.

-       Line 250: what is ‘paranemic DNA duplex’? This phrase is unclear to me.

-       The second paragraph of ‘fork reversal’ (Line 256-279): this part gives many details of publications one by one and it hardly overviews role of each molecules or how these molecules are coordinated (SMARCAL1, ZRANB3 and HLTF, etc.). Is it possible to summarise whether these molecules collaborate or how these play distinct roles in the replication fork?

-       Line 322: should ‘fork stalling’ be replication blockage or other words which means block of DNA synthesis by DNA polymerase.

-       Line 327: I do not think that 8-oxo-dG is ‘bulky lesion’. Rather 8-oxo-dG is one of the lesions which does not block DNA polymerase (or does not distort the double helix structure) and considered as non-bulky lesions.

-       The paragraph of line 412-428: this paragraph also describes many details of publications one by one but barely summarizes the content. 

-       The paragraph of line 640-650: this part is hard for me to understand.

-       The section 5 ‘Checkpoint inhibitors for cancer therapy’: this section largely describes details of clinical trial of each inhibitor and does not fit with other parts. I do not think this section is necessary to be included in this review article.  

Author Response

This review article focuses on how ATR-dependent checkpoint pathways are coordinated with the mechanism of DNA replication. In this content, author included details of the regulation of DNA replication (initiation and the roles of many components in the replisome) as well as the response of replication forks to various replication stress as well as DNA damage. I think this in-depth content will update reader’s understanding of the mechanisms of DNA replication and its ‘adaptive mechanisms’.

I listed some specific points which is unclear or can be improved (listed below).

-       The second paragraph of ‘1, initiation of DNA replication’ (line 40-55): this part is based on the research in budding yeast. I suggest authors to mention this point.

Budding yeast is mentioned in the text.

-       Recent paper also mentioned redundant roles of CDC7 and CDK1 during G1/S transition in mammalian cells (https://doi.org/10.1038/s41586-022-04698-x). This research could be incorporated in this part (or somewhere relevant).

This paper is cited in the section 1.

-       Line 103 & 115: I suggest to clarify what the pre-initiation complex (or what proteins consist of this complex). Reference: https://doi.org/10.15252/embr.201744206  

     This paper is cited to clarify the composition of the pre-IC complex.

-       Line 124 ‘replication fork stalling results in a large stretch of ssDNA accumulation’: I do not think fork stalling itself causes a stretch of ssDNA.

      The phrase is now changed to “uncoupling of replisome activity”.

-       Line 142-146 ‘This unique configuration ~ MCM central pore’: These sentences are unclear to me. Is it possible to clarify this part with figure 2?

      This is edited to better explain the importance of the FPC positioning ahead of the replisome in fork progression and strand separation.

-       Line 206-208 ‘Perhaps ~ TIM may ~ suppress R-loop~’. are there any evidences? If not, I suggest authors to remove this sentence.

      This sentence is removed as suggested.

-       Line 250: what is ‘paranemic DNA duplex’? This phrase is unclear to me.

      An explanation for the paranemic DNA duplex is included in the text.

-       The second paragraph of ‘fork reversal’ (Line 256-279): this part gives many details of publications one by one and it hardly overviews role of each molecules or how these molecules are coordinated (SMARCAL1, ZRANB3 and HLTF, etc.). Is it possible to summarise whether these molecules collaborate or how these play distinct roles in the replication fork?

      Additional discussion is added to speculate the specific roles of three distinct enzymes.

-       Line 322: should ‘fork stalling’ be replication blockage or other words which means block of DNA synthesis by DNA polymerase.

      It is changed to “replication blockage”.

-       Line 327: I do not think that 8-oxo-dG is ‘bulky lesion’. Rather 8-oxo-dG is one of the lesions which does not block DNA polymerase (or does not distort the double helix structure) and considered as non-bulky lesions.

      “Bulky” is removed.

-       The paragraph of line 412-428: this paragraph also describes many details of publications one by one but barely summarizes the content. 

      Summary statements are included to highlight the content of the paragraph.

-       The paragraph of line 640-650: this part is hard for me to understand.

      This text is rephrased to better explain the cited work.

-       The section 5 ‘Checkpoint inhibitors for cancer therapy’: this section largely describes details of clinical trial of each inhibitor and does not fit with other parts. I do not think this section is necessary to be included in this review article.  

      Thank you for your suggestion. We agree that this section may not be directly connected to the description of the basic research on DNA damage tolerance and checkpoint responses. However, we would like to provide examples of how the concept of DNA replication catastrophe is being utilized in real world. Therefore, we believe that this section is valuable as a reference of checkpoint inhibitors currently under development, which is extended from the ATR-CHK1 signaling that is discussed in the previous sections.

Round 2

Reviewer 2 Report

The authors adequately responded to all the comment I listed.